# Traffic Sign Recognition with Deep Learning: Vegetation Occlusion Detection in Brazilian Environments

**DOI:** 10.3390/s23135919

**Published:** 2023-06-26

**Authors:** Vanessa Dalborgo, Thiago B. Murari, Vinicius S. Madureira, João Gabriel L. Moraes, Vitor Magno O. S. Bezerra, Filipe Q. Santos, Alexandre Silva, Roberto L. S. Monteiro

**Affiliations:** 1Computational Modeling and Industrial Technology Program, SENAI CIMATEC, Salvador 41650-010, Brazil; thiago.murari@fieb.org.br (T.B.M.); roberto.monteiro@fieb.org.br (R.L.S.M.); 2Industrial Management and Technology Program, SENAI CIMATEC, Salvador 41650-010, Brazil; 3Institute of Science, Innovation and Technology of the State of Bahia (INCITE)—Industry 4.0, SENAI CIMATEC, Salvador 41650-010, Brazil; 4Electrical Engineering Program, College of Ilhéus, Ilhéus 45655-120, Brazil; 5Electrical Engineering Department, Federal University of Sergipe, São Cristovão 49100-000, Brazil; 6Department of Engineering and Computing, State University of Santa Cruz, Ilhéus 45662-900, Brazil

**Keywords:** object recognition, Neural Networks, YOLO

## Abstract

Traffic Sign Recognition (TSR) is one of the many utilities made possible by embedded systems with internet connections. Through the usage of vehicular cameras, it’s possible to capture and classify traffic signs in real time with Artificial Intelligence (AI), more specifically, Convolutional Neural Networks (CNNs) based techniques. This article discusses the implementation of such TSR systems, and the building process of datasets for AI training. Such datasets include a brand new class to be used in TSR, vegetation occlusion. The results show that this approach is useful in making traffic sign maintenance faster since this application turns vehicles into moving sensors in that context. Leaning on the proposed technique, identified irregularities in traffic signs can be reported to a responsible body so they will eventually be fixed, contributing to a safer traffic environment. This paper also discusses the usage and performance of different YOLO models according to our case studies.

## 1. Introduction

The automotive industry has invested in technological systems to increase safety in traffic environments. According to [1], a vehicle can be classified based on the level of automation present in an application named Advanced Driver Assistance Systems (ADAS), which is designed to use cameras, radars, embedded computer systems with internet connectivity, and other technologies to provide car features that assist the driver or even automatically control the vehicle in tasks such as parking, breaking and recognizing traffic signs. Among the various functions present in the ADAS context, traffic sign recognition (TSR) is one of the most important features since traffic signs provide information for a safe traffic environment and assist both human and autonomous driving [2].

Many TSR studies involving Artificial Intelligence (AI) methods have been recently reported in the literature. In [3,4], the detection step is based on sign colors and shapes, while the classes are predicted by a Convolutional Neural Network (CNN). However, according to [5,6,7], this approach is not reliable in terms of TSR challenges such as occlusion, which is the focus in this paper. On the other hand, well known deep learning models, such as Faster R-CNN [8], You Only Look Once (YOLO) [9], and Single Shot Detector (SSD) [10] are investigated in [5,6,11]. These models provide both object detection and classification (recognition) as their final results, being more stable than the other aforementioned detection approaches [5,7] and allowing wide possibilities in object recognition once a dataset with images and coordinates of the objects to be recognized is available.

There are well known and publicly available datasets for traffic sign detection and classification. Some of them are the Germans Traffic Signs Datasets benchmarks for classification (GTSRB) [12] and detection (GTSDB) [13], the Belgian Traffic Signs Dataset (BTSD) [14] and the Dataset of Italian Traffic Signs (DITS) [15]. However, there is a need to create datasets with more complex scenarios, since existing datasets are saturated with CNN methods achieving over 90% accuracy [16]. In addition, some of the traffic sign types in these European datasets do not exist or have different patterns in other countries such as Brazil. This is one of the factors that reinforces the need for a benchmark dataset for each country or region, as discussed by [17].

On real roads, TSR is challenged by a variety of factors, such as occlusion, illumination changes, fading and weather conditions [17,18,19]. Among these phenomena, occlusion needs special attention. The presence of obstacles in front of a sign might make it difficult to recognize because of the reduced visibility. This is a very common problem that can be caused by pedestrians, trees, other vehicles etc., [20]. Previous systems present relevant solutions for this issue, but have a very limited scope [19]. For example, when a traffic sign has a high level of occlusion, it may be unrecognizable for both humans and technological systems, thus leading to unsafe traffic.

A special case of occlusion is caused by vegetation, and its maintenance depends on the information provided by citizens or periodic checks carried out by the responsible body, e.g., Municipal Maintenance Department or a private maintenance agency. This is a harder task to commit in large cities, highways and rural areas. Thus, in the research presented herein it is investigated, the use of the YOLOv5 architecture [21] to recognize traffic signs in Brazil, evaluating is investigated, and its effectiveness in classifying traffic signs occluded by vegetation is evaluated.

The implication of the occlusion caused by vegetation on the detection of different types of objects has been, whether explicitly or not, covered by researchers. In [22], the authors show that they can reach top performance in traffic sign detection by using the integral channel features approach. Considering the datasets considered in [22], the BTSD contains cases such as occluded (including by vegetation) and damaged traffic signs, which are called challenging samples by the authors since the resulting detector’s few misses are mostly respective to them. In [23], occlusion caused by vegetation also poses a challenge associated with computer vision when the authors propose a pipeline for the detection and classification of traffic signs.

While the vegetation issue emphasized in this article is real and relevant, it is only part of all the possible variables that can influence the detection and classification of objects. Different types of occlusions, as well as weather and illuminance variance in dataset samples, have caught the attention of many researchers, as in [24,25], both oriented to road sign recognition. In light of the insufficiency of TSR applications in harsh environments, the authors in [11] propose two benchmarks focused on ice and snow environments, the Ice Environment Traffic Sign Detection Benchmark (ITSDB) and the Ice Environment Traffic Sign Recognition Benchmark (ITSRB), including numerous images with occluded traffic sign instances, yet they also propose an attention network that performs relatively well in comparison to other state-of-art networks. Some authors choose to test novel approaches in challenging datasets, such as the GTSRB, which is widely used for containing heterogeneous illumination, occlusion, etc. In [26], GTSRB is one the datasets applied to test the sliding windows approach, which, unlike the traditional machine learning (ML) techniques, considers a sequence of images to commit classification.

In the context of AV, some papers investigate the occlusion of objects in a rather generic way, dealing with this issue from traffic sign detection to pedestrian and vehicle detection. The autonomous driving environment tends to be particularly complex since the different types of objects of interest create many specific scenarios a model should specialize in. In addition to the occlusion caused by static objects (such as lampposts and buildings), target objects might interocclude or even self-occlude, as analyzed in [27]. Ref. [28] discussed how previous methods are prone to be affected by occlusions, and then a novel deep learning-based solution was proposed and posed to be robust to those scenarios, outperforming the well-established Faster R-CNN. The authors in [29], on the other hand, create a whole new dataset containing various occluded instances, studying the issue, and analyzing the detection capabilities of state-of-art architectures.

In [30] proposed a ray-tracing-based method, made possible by the inputs of a highly detailed 3D city model and the as-is situation provided by 3D mobile laser scanning (MLS), to commit the detection of occlusion of traffic lights and traffic signs caused by vegetation. The MLS is mainly responsible for capturing details of irregularly shaped objects such as vegetation (including leaves and branches, in this case) by using a light detection and ranging (LiDAR) sensor, thus complementing the information in the 3D model: the location and orientation of streets, traffic lights, and traffic signs. A voxelization of the space supplies a way to generate an occupancy grid, explored by the ray-tracing technique to localize occlusions in the 3D environment. Although having similar purposes to this article’s, this approach is limited in the sense that a 3D city model must be available to apply the proposed method. Furthermore, the accuracy and completeness of that model can easily impact the performance of this application. In contrast, this article’s ML proposed approach focuses on the occlusion of traffic signs only, and it does not lean on 3D modeling, being independent of external assistance once the training of the network is finished. In other words, it is only dependent on the input of images (captured by vehicular cameras), consisting of a 2D application.

Building of Brazilian traffic sign datasets has been reported in [4,31], at least for model testing, but as far as we know, there is no Brazilian dataset publicly available. Taking into account this limitation and the above discussion regarding vegetation occlusion, the main contributions of the this paper are summarized as follows:Development of a new Brazilian TSR dataset (BRTSD) which contains not only annotations of visible traffic signs, but also vegetation occlusion as a new class;Vegetation occlusion recognition with the use of YOLOv5 architecture.

After this section, the paper is organized as follows. The next Section 2, follows the methodology used to tackle the challenges of creating such a system, the mapping of defects and the creation of models to simulate defects for the improvement of training data. Afterwards, Section 3 presents the results obtained by the research and implementation of the system, showing its performance and effectiveness in detecting and classifying obstructed and faded traffic signs. Section 4 presents general discussions about the proposed dataset and related papers with occlusion and vegetation. Next, the conclusion in Section 5 present the discussion by asserting over the obtained results and stipulating goals for future work. Finally, a bibliographic glossary containing all the references used for research and development of the system was created.

## 2. Materials and Methods

This section presents the materials and methods used to create the TSR dataset and detector.

### 2.1. Data Set Definition

According to the Brazilian Traffic Signaling Standards (BTSS) [32], traffic regulation signs inform road users of the conditions, prohibitions, obligations, or restrictions on the use of urban and rural roads. Moreover, warning signs alert road users to potentially dangerous conditions, obstacles, and restrictions on the road or adjacent to it, indicating the nature of these situations ahead, whether permanent or intermittent. Ignoring these signs therefore is a violation of the Brazilian Traffic Code.

There are more than 160 types of traffic signs scattered across the country’s streets, roads, and highways [32]. We limited our workflow to operate with 16 main traffic sign classes, according to the object availability in the KartaView platform during the dataset creation period, where each class has at least 20 annotation instances. The classes are: No Stopping or Parking, No Parking, No Turning Left, No Turning Right, No U-turn Left, Roundabout Ahead, Right Turn or Straight only, Brazilian Stop Sign, Speed Bump Ahead, Pedestrian Crossing ahead, Roundabout Ahead, Traffic Lights Ahead, 30 km/h Speed Limit, 40 km/h Speed Limit, 50 km/h Speed Limit and 60 km/h Speed Limit. Pictures of each traffic sign type are shown in Figure 1.

Unlike the usual traffic sign datasets, BRTSD presents a new class called vegetation, attributed to traffic signs occluded by vegetation. The criteria used for the annotations of this class are part of the plaque being visible even when obstructed. Figure 2 shows images present in the dataset. From Figure 2a–c it is easy to determine we’re dealing with No Parking signs, despite the obstruction. The reason why those are classified as vegetation, even though the identified vegetation causing occlusion is rather sparse from the shown point of view, is that, from other perspectives, it might also be considerably dense and make those signs unrecognizable [30]. Moreover, detecting such a case would make it possible to anticipate maintenance, therefore avoiding the vegetation growing further and occluding the sign completely. Being able to identify partially occluded traffic signs, therefore, constitutes the ideal scenario for the proposed method. In other situations, the vegetation density is so high that the annotated signs are barely recognizable.

After this definition we followed the flowchart in Figure 3 which shows the steps taken to obtain a dataset for TSR, including KartaView API (obtaining images), automatic annotations, dataset curation, dataset export, model training and evaluation. These steps are explained in the following subsection.

### 2.2. Obtaining Images

To create a dataset for TSR, traffic images containing objects from the classes mentioned above are needed. The images presented in the proposed BRTSD consist of a set of videos recorded by a smartphone attached to a car dashboard, and a set of images obtained through the KartaView platform [33], which is an open-source street level image platform that has many traffic images of roads and streets around the world. Images corresponding to Brazilian selected tracks are downloaded through the KartaView API in the database.

### 2.3. Annotation Process

For object detection tasks such as the one presented in this work, the coordinates of each object of interest in each image are needed to train the models. Usually, these coordinates are obtained through software such as Label Studio [34], Labelme [35] and CVAT (Computer Vision Annotation Tool) [36] by drawing bounding boxes around the image objects and assigning a label to them, as shown in Figure 4.

The annotation process occurs in two stages. Firstly, a pre-trained model is used to generate automatic annotations. This stage is dedicated to lowering human labor. Correcting pre-annotated datasets is quicker and more reliable than annotating all images from the start. The model used for pre–annotation is replaced at every new iteration. As every annotation batch expands the dataset, the model evolves.

The second stage, annotation improvement, is made with the software CVAT to correct the class errors and bounding box mismatches made by the previous stage AI.

### 2.4. Model Training

One of the most suitable CNN architectures for real-time use is You Only Look Once (YOLO) [6]. Systems developed with YOLO can provide an accuracy greater than 80% while using approximately 30 frames per second, [5,6]. YOLO is an algorithm based on regression, also known as a single-stage detection algorithm [6]. The algorithm needs to normalize the input image to a fixed size, with the standard size being (608 × 608 × 3). After normalization, the input image will flow through the three layers of the architecture: a backbone layer, a neck layer, and a prediction layer. In the backbone layer, the input image height and width are compressed through a slicing operation to carry out the integration of image dimensional information into channel information [6]. The neck layer adopts a network architecture that combines the Feature Pyramid Network (FPN) and the Path Aggregation Network (PAN) structures. In this layer, information is transferred and fused to obtain predicted feature maps, with FPN working from top to bottom and PAN from bottom to top [6]. The prediction layer makes use of image features to generate and adjust bounding boxes and to predict the categories of the detected objects.

The standard YOLOv5 architectures, listed in Table 1 with their number of parameters, are trained with BRTSD. The training approach is to fine-tune YOLOv5 these architectures pre–trained on the COCO dataset [21].

First, the yolov5m model was trained for 1200 epochs. The validation losses on bounding boxes, objectness and class are presented in Figure 5 for this training. It is possible to verify that the losses are minimum between 500 and 600 epochs, which implies that training models for more than 600 epochs does not reflect improvements in the model performance. Therefore, all models in Table 1 are trained for 600 epochs and compared to each other. Each training is performed with YOLOv5 default hyperparameters.

### 2.5. Evaluation Metrics

The metrics used to compare the obtained models are precision, recall, F1-score, average precision (AP) and mean average precision (mAP). All these metrics take into account that a true positive (TP) prediction is characterized by correct bounding box coordinates and class attribution of an existing object of interest, otherwise a false negative (FN), and/or false positive (FP) is/are generated. Each FN is a ground truth (GT) object that has no correct prediction match, while FPs are predictions that do not have correct GT matches. True Negatives (TN) are not observed because they represent all bounding boxes that do not contain any object of interest in an image, and there are too many of them [37].

A prediction bounding box is correct when its coordinates are close to the GT bounding box coordinates. In the object detection tasks this condition is commonly evaluated through the intersection over union (IoU) of the predicted and GT bounding boxes, as represented by (Equation 1), where $B_{gt}$ is the ground truth bounding box and $B_{p}$ is the predicted bounding box.
(1)$$IoU=\frac{area(B_{gt}\cap B_{p})}{area(B_{gt}\cup B_{p})}$$

For each predicted object, the detector provides a confidence score, which shows how sure the model is that a bounding box contains an object of a specific class. If IoU is greater than an IoU threshold and the predicted class matches the GT class with a confidence score greater than a threshold, a TP is found.

Precision is calculated through (Equation 2) and indicates the degree of how precise the model is attributing classes to detected objects over the evaluation samples. Recall is given by (Equation 3) and evaluates if the model finds all the objects of interest. The F1-score is the harmonic mean of precision and recall, computed by (Equation 4). AP is calculated according to Common Objects in Context (COCO) style-evaluation through the FiftyOne library by computing matches for 10 IoU thresholds from 0.5 to 0.95 with steps of 0.05 and averaging (precision, recall) points over the confidence scores [38].
(2)$$Precision=\frac{TP}{TP+FP}$$
(3)$$Recall=\frac{TP}{TP+FN}$$
(4)$$F_{1}=2\times \frac{precision\times recall}{precision+recall}$$

Precision, recall, F1-score, and AP are computed for each class. mAP is the average of AP values over all classes, and all these metrics are calculated using the FiftyOne library.

The hardware used to make predictions has with Intel(R) Core(TM) i5-7200U CPU @ 2.50 GHz CPU, 16 GB RAM, and an NVIDIA GeForce 920MX GPU (Note that we are not recommending this hardware for TSR application. These specifications are presented for reference only. Refer to [5,6] for results with better hardware).

## 3. Results

This section presents the results and discussions regarding the specifications of the created dataset. Additionally, YOLOv5 training and validation results are discussed.

### 3.1. Dataset

Table 2, shows a comparison between our dataset and the European datasets for detection (BTSD, DITS and GTSDB). The major limitation of our dataset when compared with BTSD and GTSDB is the number of classes, which is also not close to the more than 160 classes present in BTSS. However, our dataset outperforms GTSDB and DITS in terms of image and object counts. In addition, with the presented methodology and its tools it can be expanded faster.

The objects per class distribution are presented in Figure 6. The classes do not present an uniform distribution because to the fact that specific signs, such as No Parking, No Parking or Stopping and 50 km/h Speed Limit present a much higher frequency among the analyzed images, while, traffic signs such as Roundabout Ahead, and Traffic Lights Ahead, in contrast, are relatively scarce. Due to this low presence of some traffic sign classes on the roads and streets, it is difficult to achieve a dataset with all classes and a balanced distribution. It is also observed in other traffic sign datasets as presented in [4,5,12].

### 3.2. Training Results

Figure 7 shows the training and validation loss for the models in Table 1 for 600 epochs. The training loss is lower or close to the validation loss for every model and this number of epochs is good enough to train on the obtained dataset, because these validation losses ended up presenting the same behavior shown in Figure 5.

### 3.3. All Classes Validation Results

Table 3 presents the mAP, precision and recall for models trained on BRTSD. It can be noted that comparing YOLOv5 models, the mAP increases as the architectures becomes large, and yolov5x presents the higher value. However, yolov5m should be a better choice to use because it has fewer parameters than yolov5x and has a mAP, precision and recall close to the obtained with this one.

### 3.4. Vegetation Class Validation Results

Table 4 presents the metrics for each YOLOv5 model regarding only the vegetation class, which has 60 validation annotations (support). These results refer to 0.25 confidence threshold detections. Note that the model yolov5m has a better F1-score with the highest precision, although the yolov5x model finds more objects with better recall. The AP increases as the model has more parameters as shown in Table 1.

Since vegetation occlusion can be difficult to detect and the dataset does not contain a great number of instances for this class, the confidence score is analyzed. Figure 8a shows the TP confidence values (circles), mean (triangle), and standard deviation behavior (lines) of each model for vegetation class. Figure 8b presents the same for FP predictions. It can be noted that all models can achieve highly confident TP predictions with averages from 0.702 to 0.771. However, some FP predictions are highalso highly confident too, being the, with confidence averages in the range of 0.4 to 0.56.

Figure 9 shows model mistakes (FP and FN) involving vegetation class. In Figure 9a,b, it is noticeable that the model might, mistakenly, detect patterns such as signs (highlighted by the red boxes) immersed in dark environments. Additionally, FP cases are are observed, as presented in Figure 9c–e where the trained model apparently detects reddish objects as if they are stop signs occluded by vegetation. Finally, Figure 9f shows that vegetation class FNs may occur if traffic signs with a low level of vegetation occlusion are labeled as vegetation class as explained, in Section 2.1.

## 4. Discussion

It is difficult making to make a fair comparison between the models obtained in this paper and those presented in other works, since a new dataset is proposed, with different classes, image resolutionresolutions, class distributiondistributions, and so on. Recently, ref. [19] presented a similar study involving occlusion in TSR, but usingused their own dataset. Authors haveThe authors achieved precision and recall metrics of 0.81 and 0.79, respectively. In Table 3, the proposed yolov5m model, for example, has 0.92 precision and 0.85 recall. However, the dataset described in [19] might have more instances with challenging situations.

Additionally, ref. [19], it is presented 0.8 to 0.83 accuracy results regarding the classification of traffic signs occluded by vegetation. However, accuracy relies on TNsTN computation, which is not observed in the context of yolov5the YOLOv5 architecture, as mentioned before. In contrast, Table 4 presents precision, recall and f1-score for this specific occlusion class, and it is noticeable that the major problem in the presented models are FNs like such as the one shown in Figure 9e, since recall values are lower than precision onesvalues in general.

Analyzing the mAP metric regarding the general TSR task, in [5], the application of Faster R-CNN in GTSDB achieves a mAP of 0.911, while the yolov5m model, in Table 3, reaches 0.718. However, the GTSDB dataset has no traffic signs occluded by vegetation, so the respective Faster R-CNN model is not tested in that situation, consequently resulting in a better mAP. Another Faster R-CNN model trained in ITSDB [11], which considers occlusion and nightly scenarios, has an AP of 0.456. Therefore, it is evident that, the more realistic and challenging the dataset is, the worse the performance of trained models tends to be.

In fact, traffic signs with a high level of occlusion are not well recognized by both humans and AI. As discussed above, it can possibly drop the metrics of a model if it is evaluated in those situations. The solution proposed is to anticipate the maintenance and prevent such cases, while traffic signs are still partially visible and exposed for detection. From the results presented in Table 4 and Figure 8, even though there is a small number of instances for the vegetation class in the presented dataset, the precision, recall, and confidence for the resulting models show that the inclusion of this class in TSR datasets can be useful to make this kind of information available for street and road maintainers. Since it is a traffic safety issue, lives can be saved even if one single sign occluded by vegetation is reported through the vehicle’s ADAS to the responsible body, e.g., Municipal Maintenance Department or a private maintenance agency.

Regarding the model mistakes in Figure 9, improvements can be made with the addition of night-time annotations to the dataset to mitigate cases in Figure 9a,b. For now, establishing a limit time for vegetation classification should be a solution. Regarding Figure 9c–e, in which the model mistakes red objects behind the vegetation, adding background images (without traffic signs) to the dataset, as suggested in [14], may solve this scenario. In the latter case, Figure 9e, although the predicted class does not match the annotated vegetation class, it is not completely wrong because the correct information about the speed limit is obtained.

## 5. Conclusions and Future Work

This article, developed for the needs of the identification and classification of traffic signs, showed a significant level of efficiency, while probing the accuracy of vegetation occlusion detection based on a private dataset created for this study (BRTSD). Hence, in this paper, it is used the version 6.2.3 of the series of YOLOv5, to evaluate its performance. It is possible to identify which model had better performance. It is verified that the yolov5n model has superior detection speed, while yolov5x had a better mAP. Since yolov5m has better detection speed than yolov5x with a similar mAP and higher precision (92.9%) for vegetation occlusion detection, we concluded that architecture is a better choice.

The model was trained, validated, and tested on a specially designed traffic sign database consisting of 16 different classes, 1 defect class (vegetation occlusion) with 5631 images and 8065 annotations. We identified false positive detections for vegetation class due to few images into authors private data set.

In the future work, the authors will further optimize the performance increasing the dataset and stability of our detection model. We are studying other types of traffic signals defects as fading, graffite and others defects to turn our attention to how to process detection tasks with large amounts of data in real time. Also, we are studying to increase our private dataset including data argumentation methods.

## Figures and Tables

**Figure 1 sensors-23-05919-f001:**
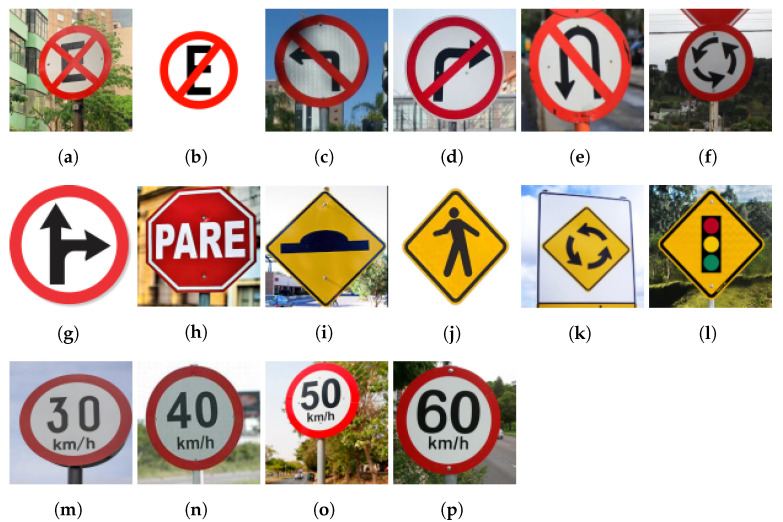
(**a**) No Stopping or Parking. (**b**) No Parking. (**c**) No Turning Left. (**d**) No Turning Right. (**e**) No U-turn Left (**f**) Roundabout Ahead. (**g**) Right Turn or Straight Only. (**h**) Brazilian Stop. (**i**) Speed Bump Ahead. (**j**) Pedestrian Crossing ahead. (**k**) Roundabout Ahead. (**l**) Traffic Lights Ahead. (**m**) 30 km/h Speed Limit. (**n**) 40 km/h Speed Limit. (**o**) 50 km/h Speed Limit. (**p**) 60 km/h Speed Limit.

**Figure 2 sensors-23-05919-f002:**
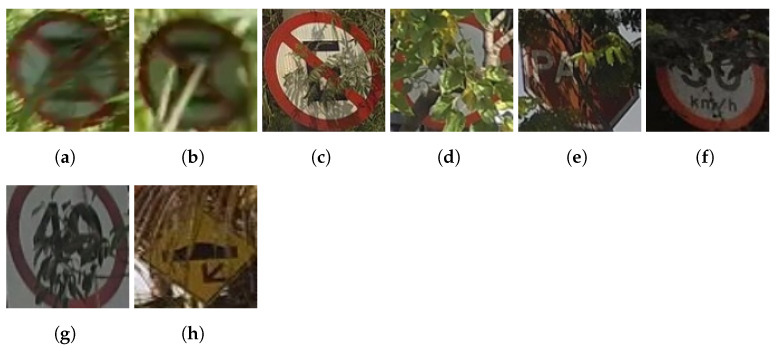
From (**a**–**h**) Images depicting occlusion by vegetation.

**Figure 3 sensors-23-05919-f003:**
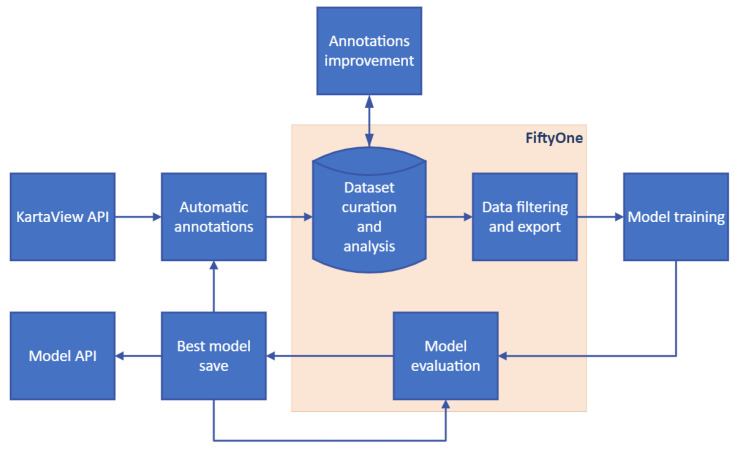
Flowchart.

**Figure 4 sensors-23-05919-f004:**
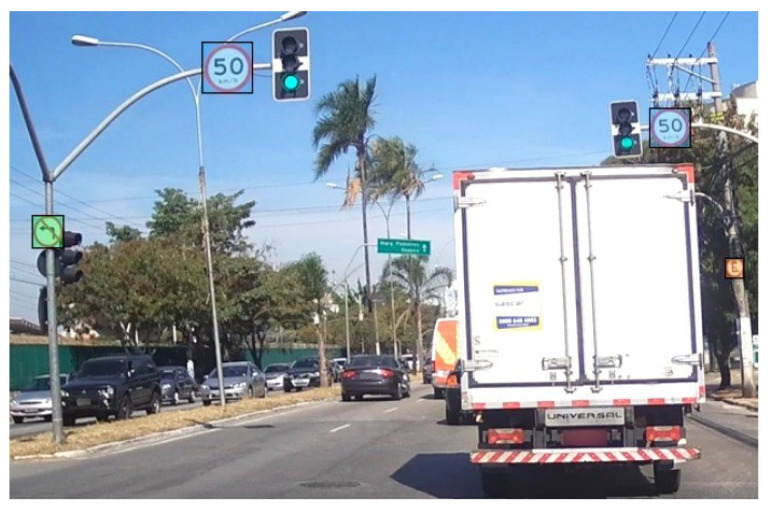
Image annotation.

**Figure 5 sensors-23-05919-f005:**
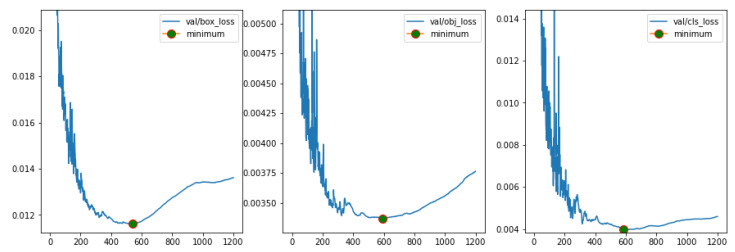
Validation loss for yolov5m trained with 1200 epochs.

**Figure 6 sensors-23-05919-f006:**
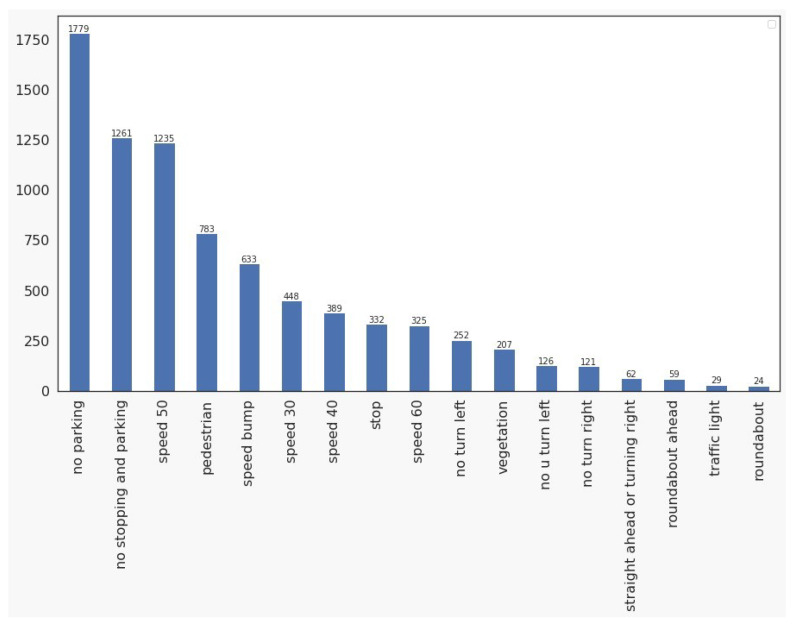
Dataset class distribution.

**Figure 7 sensors-23-05919-f007:**
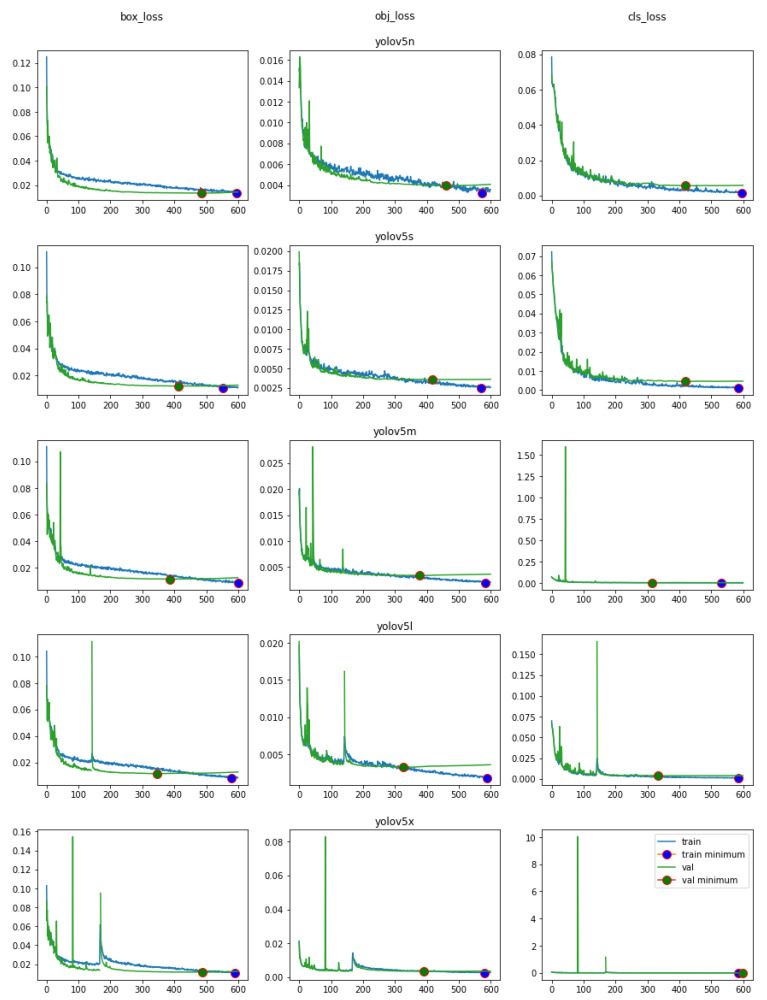
Training and validation losses for all models.

**Figure 8 sensors-23-05919-f008:**
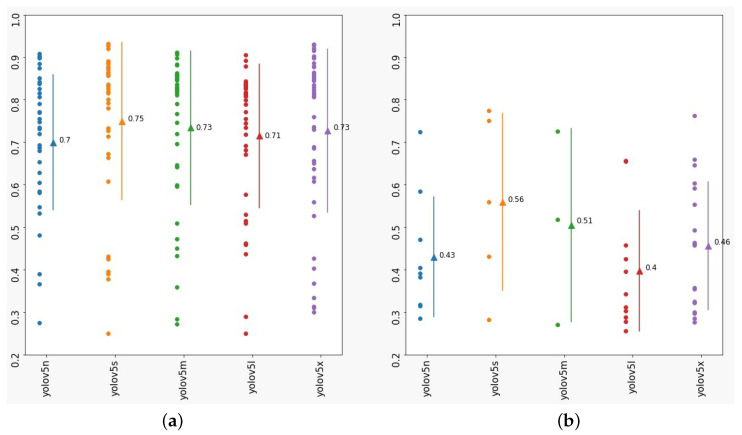
Confidence for vegetation class: (**a**) TP and (**b**) FP.

**Figure 9 sensors-23-05919-f009:**
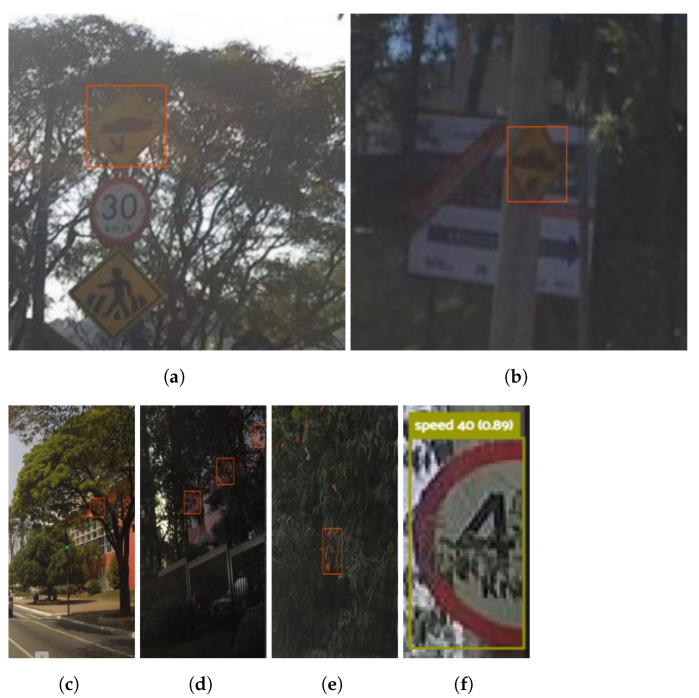
Vegetation class. (**a**–**e**) FPs, (**f**) FN.

**Table 1 sensors-23-05919-t001:** Trained models and number of parameters.

Model	Parameters (M)
yolov5n	1.9
yolov5s	7.2
yolov5m	21.2
yolov5l	46.5
yolov5x	86.5

**Table 2 sensors-23-05919-t002:** Object detection datasets.

Dataset	Classes	Images	Objects
BTSD	62	9006	13,444
DITS	3	1872	1891
GTSDB	43	900	1206
ITSDB	1	5806	43,290
BRTSD	17	5631	8065

**Table 3 sensors-23-05919-t003:** mAP for all class detection validation.

Model	mAP	Precision	Recall
yolov5n	0.609	0.810	0.832
yolov5s	0.672	0.856	0.869
yolov5m	0.718	0.921	0.851
yolov5l	0.700	0.889	0.893
yolov5x	0.750	0.951	0.865

**Table 4 sensors-23-05919-t004:** Metrics for vegetation class detection.

Model	Precision	Recall	F1-Score	AP
yolov5n	0.795	0.583	0.673	0.337
yolov5s	0.878	0.600	0.713	0.379
yolov5m	0.929	0.650	0.765	0.427
yolov5l	0.771	0.617	0.685	0.411
yolov5x	0.726	0.750	0.738	0.454

## Data Availability

The data are not publicly available due to further project application.

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
