# Peer review of "Traffic Sign Recognition with Deep Learning: Vegetation Occlusion Detection in Brazilian Environments"

_sensors, 2023, doi:10.3390/s23135919_

Round 1
Reviewer 1 Report
The authors in the work developed their own dataset containing road signs with a group of signs obscured by vegetation. They also fine-tuned the different version yolov5 network for on their own dataset and make the results analysis.
I have a few concerns about this study
What is the efficiency of networks trained on publicly available datasets for samples with vegetation occlusion. Maybe there Is no need to create additional dataset
What is the main reason for creating the vegetation occlusion class in my opinion the main task is the traffic sign recognition not the occlusion itself because the signs could be occluded by anything ex. other cars.
Why the BTSS is not listed in the table 2
The column support in table 4 is obsolete because the value was mentioned in the text and the value is the same for all cases
Fig 9 It should be better explained why a,b,f is treated as fp
There are some typos in text example soccluded line 110
Author Response
Dear Reviewer,
We thank you for your thorough review and highly appreciate the comments and suggestions, which contributed to improving the quality of the publication. Please find below a detailed response to each of the comments.
We have also evaluated the manuscript to correct typo and and grammatical errors.
-----------------------------------
The authors in the work developed their own dataset containing road signs with a group of signs obscured by vegetation. They also fine-tuned the different version yolov5 network for on their own dataset and make the results analysis.
I have a few concerns about this study.
1. What is the efficiency of networks trained on publicly available datasets for samples with vegetation occlusion. Maybe there Is no need to create additional dataset.
Response: Public datasets do not contain Brazilian traffic signs, as explained in Section 1, lines 105-106. Furthermore, in datasets such as GTSDB, there is no annotation with respect to traffic sign quality. In our proposed dataset, the traffic signal quality condition is annotated.
2. What is the main reason for creating the vegetation occlusion class in my opinion the main task is the traffic sign recognition not the occlusion itself because the signs could be occluded by anything ex. other cars.
Response: Thank you for your observation. The objective of this manuscript is to detect and classify the failure mode in traffic signs by vegetation occlusion and to advise the timing to commit maintenance/cut vegetation by the Municipal Maintenance Department or a private maintenance agency (see Section 1, lines 49-51). It is not interesting to send temporary messages for occlusion maintenance, such as cars and/or trucks, because it would not require the intervention of the maintenance team.
3. Why the BTSS is not listed in Table 2
Response: Thank you for your comment. BTSS is not a dataset available for comparison. BTSS is a Brazilian Traffic Signaling Standards, as described in line 126.
4. The column support in table 4 is obsolete because the value was mentioned in the text and the value is the same for all cases
Response: Thank you for your suggestion. We removed column support in Table 4.
5. Fig 9 It should be better explained why a,b,f is treated as fp
Response: An explanation has been added in Section 2.5, lines 208-210. The manuscript was updated as follows: “Each FN is a ground truth (GT) object that has no correct prediction match, while FPs are predictions that do not have correct GT matches.”
Reviewer 2 Report
Introduction:
Contributions: development of dataset is valid, but not enough for Q1 journal. "Study of vegetation..." is not contribution by itself. You should rephrase this contribution.
Dataset
"For simplification purposes, we limited our workflow to operate with 16 main traffic sign classes" 16 of 160? Is it too small for the database? It should be only beginning of the dataset creation. Final version should have all signs or it is not fully useful. I know that it is difficult to distinguish all 160 signs, but what is the criteria for choosing one over another? Have you considered actual ANN's (YOLO) work - how does it operate? Is the ANN recognize the shape of signs, size, letters, etc or combination?
Lines 139-150: I don't understand why do you not classify recognizable signs within their classes, not as vegetation? Is it the point to recognize as much as possible? Only unrecognizable should be classify as vegetation. Than, new algorithm should be use to try to detect occluded signs.
You should increase a number of signs included in your dataset.
You should reorganize the paper to separate dataset from the detection problem with vegetation occlusions.
The paper is between reject and major revision. The proposed dataset has several deficiencies.
Author Response
Dear Reviewer,
We thank you for your thorough review and highly appreciate the comments and suggestions, which contributed to improving the quality of the publication. Please find below a detailed response to each of the comments.
We have also evaluated the manuscript to correct typo errors.
--------------------------
1. Introduction:
1.1 Contributions: development of dataset is valid, but not enough for Q1 journal. "Study of vegetation..." is not contribution by itself. You should rephrase this contribution.
Response: Thank you for your comment. The objective of this manuscript is to detect and classify failure modes in traffic signs by vegetation occlusion and to advise the timing of maintenance/cut vegetation by the Municipal Maintenance Department. In our proposed dataset, the traffic sign vegetation condition is annotated with longitude and latitude information, while the public datasets do not have this information. Updated in the manuscript, Section 1, lines 49-51 and Section 4, lines 309-312. Title update according your comment.
2. Dataset
2.1 "For simplification purposes, we limited our workflow to operate with 16 main traffic sign classes" 16 of 160? Is it too small for the database?
Response: We limited the traffic signs because 16 classes were enough to detect the vegetation class with a high quantity of instances. The explanation has been added in the revised article, Section 2.1, lines 133-135.
2.2 It should be only beginning of the dataset creation. Final version should have all signs or it is not fully useful.
Response: Our research is about detecting and classifying the failure mode in traffic signs by vegetation occlusion, so the most important part is to detect this failure mode. The objective was accomplished using only 16 classes to detect traffic sign occlusion by vegetation.
2.3 I know that it is difficult to distinguish all 160 signs, but what is the criteria for choosing one over another?
Response: We limited our workflow to operate with 16 main traffic sign classes, according to the availability of objects in the KartaView platform during the dataset creation period, whereas each class had to have at least 20 annotation instances. The proposed dataset has enough classes to make it possible to detect vegetation occlusion in different conditions to have good accuracy in the results. Our model can be used to complement the features available in other traffic sign classification models. In this way, it adds the functionalities of classifying the occlusion by vegetation and determining the spatial location of the traffic signs.
2.4 Have you considered actual ANN's (YOLO) work - how does it operate? Is the ANN recognize the shape of signs, size, letters, etc., or combination?
Response: Yes, we classified “good traffic signals” and “bad” occlusions by vegetation, focusing on maintenance. The fully connected part of YOLO (ANN) recognizes the shape of signs, along with other relevant features such as size, colors, and other distinguishing characteristics of the traffic signs. The combination of these features helps YOLO's ANN classify the different types of traffic signs. In our work, specifically, ANN recognizes whether the traffic signs (16 classes determined by authors) have vegetation occlusion or not. However, it can also recognize traffic sign classes if they are not occluded by vegetation. In other words, the main objective is to detect vegetation occlusion, but the developed model does more than that.
2.5 Lines 139-150: I do not understand why do you not classify recognizable signs within their classes, not as vegetation? Is it the point to recognize as much as possible? Only unrecognizable areas should be classified as vegetation. Then, a new algorithm should be used to try to detect occluded signs.
Response: The situation where we have partial vegetation occlusion, but it is possible to recognize the traffic sign the model identifies as vegetation, is explained in Section 2.1, lines 145-153.
2.6 You should increase a number of signs included in your dataset.
Response: We understand that it is ideal to increase the number of signs, but our research’s emphasis is classifying and detecting the failure mode in traffic signs by vegetation occlusion, so the most important part is to detect the failure mode in traffic sign classes, as referenced in Section 4, lines 313-320 and Section 5, lines 334-335. Therefore, the number of signs in our dataset was sufficient to reach our objectives.
2.7 You should reorganize the paper to separate dataset from the detection problem with vegetation occlusions.
Response: Thank you for your suggestion, but we would like to keep both analyses in a unique manuscript to improve the readability of the audience.
Round 2
Reviewer 1 Report
Thank you for the updates
The quality of the text is ok
Author Response
Dear Reviewer.
We truly appreciate the time and effort you dedicated to evaluate the manuscript and provide constructive suggestions.
Best Regards.
Reviewer 2 Report
Footnote 1 is not necessary. Maybe it should be removed. It is known thing.
You misunderstood my comment "You should reorganize the paper to separate dataset from the detection problem with vegetation occlusions". I meant that you should reorganize sections, not to write 2 papers.
Author Response
Dear Reviewer.
We truly appreciate the time and effort you dedicated to evaluating the manuscript and provide constructive suggestions.
--------------------
1. Footnote 1 is not necessary. Maybe it should be removed. It is known thing.
Response: Thank you. We agreed with your suggestion and removed the footnote 1 from the manuscript.
2. You misunderstood my comment "You should reorganize the paper to separate dataset from the detection problem with vegetation occlusions". I meant that you should reorganize sections, not to write 2 papers.
Response: We apologize for any confusion caused by our previous response. Upon reviewing your comment, we now understand that you recommended reorganizing sections of the paper rather than suggesting the creation of two separate papers. We would like to assure you that we have carefully considered your feedback. We believe that the latest revision has significantly improved the clarity of the paper and facilitated better readability for the readers. The Materials and Methods section, as well as the Results section, present the dataset and detection problem separately.
Best Regards.